# Peer review of "Tetrahedral DNA Framework-Programmed Electrochemical Biosenors with Gold Nanoparticles for Ultrasensitive Cell-Free DNA Detection"

_nanomaterials, 2022, doi:10.3390/nano12040666_

Round 1
Reviewer 1 Report
The manuscript nanomaterials-1578527 “Tetrahedral DNA Framework-Programmed Electrochemical Biosenors with Gold Nanoparticles for Ultrasensitive Cell-free DNA Detection” for Nanomaterials proposed a new approach for an ultrasensitive electrochemical biosensor based on tetrahedral DNA framework modified gold nanoparticles interface for detection of Tumor-associated cell-free DNA. The idea of the manuscript is interesting. Therefore, after carefully reading through the manuscript, this manuscript would be accepted after major revision.
The comments are as following:
It seems the Au nanoparticles are important but no experiment and its characterization data were shown in the manuscript. The authors should be added these data and experimental processes.
In Figure 1. HAuCl4 is not gold particle and why the authors though that simple dropping of HAuCl4 could form the clusters?
As the authors point out, the main cfDNA is in the blood but there is no realistic test data. I think the bovine serum would not be suitable model to confirm the clinical utility. Therefore, the authors should test the human related level of the test.
Figure 5, the authors tested the specification of the proposed DNA sensors using target, random, and mismatched DNA. The sequence and the data are important. Therefore, Table S1 should be placed in the main manuscript.
Reviewer 2 Report
The authors say: “gold nanoparticles (AuNPs) can improve the smoothness of the electrode surface”, line 57.
They should be careful and rewrite the sentence, because AuNPs are generally used to nanostructure the surface, which leads to roughening of the surface. The TEM images in Figure 1 show the opposite. Reference 13 is not a good example if the authors want to show that the surface is smoothed with a nanomaterial such as AuNPs.
The representation of the star-shaped gold nanoparticles is confusing to the reader. To try to make it as similar as possible to reality, they should be spherical.
When the authors describe the formation of AuNPs in section 3.2, they should indicate the diameter of the AuNPs. As well as the expected diameter in the formation of 3d DNA structures. So that the reader can have a clearer idea of ​​the relationship on the surface from the beginning, since only the expected size of the 3D structures appears but at the end of the manuscript.
How do the authors avoid nonspecific interactions if they do not coat the AuNPs with a thiol or other molecule? How do they explain it?
The authors should explain the evidence that they obtain pyramid-shaped 3D structures like the one they show in their drawings and not other possible configurations.
The authors should explain at what potential the chronoamperometry shown in Figure 3a is performed and how they obtain the current intensity values ​​to represent Figure 3b. If the values ​​are obtained from the chronoamperometry, what time do they select as the appropriate one to choose the current intensity value or if they choose it from the result of cyclic voltammetry.
The authors show comments on line 205, figure S4, but they should explain why this electrochemical response with two reversible signals is due, etc…
The authors make the cyclic voltammetry measurements in the absence of O2? At negative potentials, O2 could be reduced.
What parameters have been optimized for the development of the genosensor?
The authors should include the comparison of the analytical parameters obtained with the results obtained by other authors using, for example, different techniques such as fluorescence, electrochemiluminescence or other electrochemical sensors.
Round 2
Reviewer 1 Report
The manuscript has been improved and now it is ready to be published in this journal.
Author Response
We really appreciate the reviewer’s comment and affirmation.
Reviewer 2 Report
I suggest including some more reference, such as:
“Gold nanoparticles in virus detection: Recent advances and potential considerations for SARS-CoV-2 testing development”
“Gold nanoparticles as electronic bridges for laccase-based biocathodes”.
I appreciate the new figure S4 made to justify the formation of the pyramids. But, the authors should explain based on this result the presence of the 3D pyramids. They should explain the results they obtain based on the height of the image, for example, or carry out some additional experiment, where the difference can be observed.
